# Borrowing Concurrent Information from Non-Concurrent Control to Enhance Statistical Efficiency in Platform Trials

Jialing Liu [1], Chengxing Lu [2], Ziren Jiang [1], Demissie Alemayehu [3], Lei Nie [4] and Haitao Chu [1,3,*]

1    Division of Biostatistics, University of Minnesota, Minneapolis, MI 55455, USA
2    Oncology Biometrics, AstraZeneca, Waltham, MA 02451, USA
3    Statistical Research and Data Science Center, Global Biometrics and Data Management, Pfizer Inc., New York, NY 10017, USA
4    Division of Biometric IV, Office of Biostatistics, OTS/CDER, U.S. Food and Drug Administration, Silver Spring, MD 20993, USA
*    Correspondence: chux0051@umn.edu

**Abstract:** A platform trial is a trial involving an innovative adaptive design with a single master protocol to efficiently evaluate multiple interventions. It offers flexible features such as dropping interventions for futility and adding new interventions to be evaluated during the course of a trial. Although there is a consensus that platform trials can identify beneficial interventions with fewer patients, less time, and a higher probability of success than traditional trials, there remains debate on certain issues, one of which is whether (and how) the non-concurrent control (NCC) (i.e., patients in the control group recruited prior to the new interventions) can be combined with the current control (CC) in the analysis, especially if there is a change of standard of care during the trial. Methods: In this paper, considering time-to-event endpoints under the proportional hazard model assumption, we introduce a new concept of NCC concurrent observation time (NCC COT), and propose to borrow NCC COT through left truncation. This assumes that the NCC COT and CC are comparable. If the protocol does not prohibit NCC patients to change the standard of care while on study, NCC COT and CC likely will share the same standard of care. A simulated example is provided to demonstrate the approach. Results: Using exponential distributions, the simulated example assumes that NCC COT and CC have the same hazard, and the treatment group has a lower hazard. The estimated HR comparing treatment to the pooled control group is 0.744 (95% CI 0.575, 0.962), whereas the comparison to the CC group alone is 0.755 (95% CI 0.566, 1.008), with corresponding $p$-values of 0.024 versus 0.057, respectively. This suggests that borrowing NCC COT can improve statistical efficiency when the exchangeability assumption holds. Conclusion: This article proposes an innovative approach of borrowing NCC COT to enhance statistical inference in platform trials under appropriate scenarios.

**Keywords:** platform trials; non-concurrent control; survival data analysis; master protocol; left truncation; right censoring





## 1. Introduction

Randomized controlled trials (RCTs), in which subjects are randomized at the same time as the treatment and control arms, have been considered the gold standard for the development of pharmaceuticals and medical devices [1] since their introduction in the 1930s and 1940s. However, over the years, traditional RCTs have increasingly become unable to meet the challenges of developing new therapies to treat patients with unmet medical needs. RCTs are known to be associated with high failure rates (with estimated failure rates of 36%, 68%, and 40%, respectively, for Phase I, Phase II, and Phase III studies) and high costs (with an average estimated cost $5 billion to develop a new molecule) [2,3].

With such challenges, there is increased interest in expediting late-stage drug development through master protocol trial designs that evaluate multiple drugs and/or multiple

disease subpopulations in parallel under a single protocol without the need to develop new protocols for every trial [4,5]. Understanding whether a drug is safe and effective in a clinical trial can be a very time-consuming process. As a potential solution, the master protocol design has several advantages over a traditional clinical trial design, including the ability to efficiently evaluate multiple treatments, more rapid identification of effective therapies, and improved data quality and efficiency through shared and reusable infrastructure [5]. Oncology has been the area that pioneered such designs, with numerous successful examples such as the Imatinib Target Exploration Consortium Study B2225 (the first master protocol study started in 2001), BATTLE-1/2, ALCHEMIST, LUNG-MAP, STAMPEDE [6,7], and many others conducted increasingly over the past 5 years. Recently, during the global pandemic, master protocols have again successfully brought efficiency in evaluating multiple drugs simultaneously for the treatment or prevention of COVID-19 [8,9]. With the demand from increasing applications in master protocol designs FDA has worked with clinical trial experts to release guidance for designing and implementing master protocols for Oncology and COVID-19 trials [5,10].

Master protocols may incorporate concurrent interventional investigations targeting either a single disease or multiple diseases characterized by a biomarker or disease category [11]. Master protocols typically include basket, umbrella, and platform trial designs [11]. A basket trial refers to a clinical trial design in which a targeted therapy is tested simultaneously in patient groups with diverse cancer types that have a common underlying molecular abnormality. More specifically, this approach involves putting patients with different cancer types into a basket based on their shared genetic mutation or biomarker, regardless of their specific cancer type or location in the body, allowing for a faster and more efficient evaluation of the targeted therapy's effectiveness. For example, the Rare Oncology Agnostic Research (ROAR) study evaluated the efficacy of dabrafenib and trametinib in treating rare cancers with the BRAF V600E mutation. This basket trial demonstrated that dabrafenib and trametinib significantly improved long-term survival in patients with BRAF V600E-mutant anaplastic thyroid cancer, representing a meaningful treatment option for this rare and aggressive cancer type [12,13]. Unlike the basket trial, which is aimed at a targeted therapy, an umbrella trial simultaneously evaluates multiple targeted therapies or treatment strategies for patients suffering from a specific disease or condition. The patients with a specific disease (such as a single type of cancer) are categorized into various subgroups according to their predictive biomarkers or other traits, or a combination of both. Patients within each subgroup would then be assigned a different treatment [6]. Platform trials, in particular, examine multiple treatments in the same protocol while different experimental treatments can join and leave the trial at different times. Compared to traditional RCTs, platform trials significantly improve efficiency, since they allow for frequent interim adaptations while evaluating multiple treatments at the same time with a shared common control group. This not only gives more patients the benefit of participating in clinical trials to receive promising active treatments, but also allows for faster and cheaper testing of more treatments [14]. Simulation studies [7,15] showed that in platform trials in a five-experimental-arm setting, compared to multiple traditional two-arm trials, the mean total sample size can be reduced by roughly 40% and the average drug development time can be shortened by roughly 40%.

One impactful platform trial in the oncology area is I-SPY2 for "Investigation of Series Studies to Predict Your Therapeutic Response with Imaging and Molecular Analysis". It is a neoadjuvant, adaptive, Phase II platform trial for locally advanced breast cancer enrolled initially in 2010 [16,17]. The trial is a collaboration among the National Cancer Institute, FDA, more than 20 cancer research centers, and major pharma/biotech companies. In more than a decade, the trial has evaluated more than 20 agents, screened over 3800 subjects, and enrolled more than 2000 subjects. The innovative design features include responsive adaptive randomization, frequent interim evaluations allowing for arms being stopped or "graduated" (claiming success) early, and usage of a common control. This has tremendously enabled efficiency improvement and the acceleration of drug develop-

ment processes. A more recent example of the platform trial is the MORPHEUS study conducted by a single-sponsor Roche [18]. Multiple cancer immunotherapy (CIT)-based combinations in different tumor types were evaluated within the same trial infrastructure. This global, open-label, randomized, Phase Ib/II trial enrolled patients with one of the following cancers: pancreatic ductal adenocarcinoma (PDAC), gastric or gastro-esophageal junction cancers (GC/GEJ), hormone receptor-positive or triple-negative breast cancers (HR+/TNBC), non-small cell lung cancer (NSCLC), or colorectal cancer (CRC). Within each tumor type, a sub-study was designed in which there were multiple CIT combination arms. We compared them with a single, common, standard-of-care control arm [19,20]. Another example is the RECOVERY trial. The RECOVERY platform trial rapidly identified the beneficial effects of dexamethasone in SARS-CoV-2 hospitalized COVID-19 patients while examining multiple treatments of Dexamethasone, Lopinavir-Ritonavir, Hydroxychloro-quine, Azithromycin, Tocilizumab, Convalescent Plasma, Regen-COV, Aspirin, Colchicine, or Baricitinib in preventing death in patients with COVID-19 (Figure 1). It includes multiple treatments for the main randomization, which can be performed simultaneously as appropriate with a common control group. A common control group was used as a comparator for multiple treatments in the RECOVERY trial. In feasible cases, it is better to assign a common control group for multiple treatments than have an independent control group for each treatment [8,9,14]. The benefit of this design is that it reduces the sample size and shortens the drug development time, which means it is possible to find beneficial treatments with fewer patients, reducing patient failure, decreasing time, and increasing the probability of success. Additionally, the platform trial has been increasingly used, not only in the fields of oncology and infectious diseases, but also in designing digital mental health interventions and other areas. Implementing platform trials with multiple interventions can improve global mental health care and effectively manage limited resources [14].

Although platform trials offer improved efficiency in drug development, challenges exist from the flexibility of allowing multiple treatment arms to join and leave the trial at different times. On one hand, large platform trials that have been conducted for a long period of time, such as I-SPY2 [15,21], have accumulated enriched information in the common control that can tremendously increase power if used appropriately. On the other hand, the control patients not randomized at the same time as the experimental treatment of interest (i.e., non-concurrent controls; NCC) may have different patient populations compared to the experiment arm, with a temporal shift in measured and unmeasured baseline characteristics that could be prognostic factors. In addition, even if the patient population does not change over time, the change in standard medical practice (e.g., the standard of care) or trial conduct (e.g., instruments to measure outcomes) may lead to a difference between the NCC and CC patients. For example, patients in the non-concurrent control arm may receive a different standard medical practice than those in the treatment arm due to changes in standard over time. In contrast, the concurrent control and treatment arms receive the same standard medical practice at the same time, reducing the potential for bias due to differences in standard medical practice between the two arms. Therefore, it is crucial to carefully assess the potential sources of bias and variability when involving the use of non-concurrent control group. Combining NCC and CC without careful consideration may introduce bias [22].

In a platform trial, each time a new treatment is added to a trial with existing treatment(s), the control patients recruited prior to a newly added experimental treatment are considered as non-concurrent control, whereas the control patients who are randomized concurrently with the new experimental treatment are considered as the concurrent control (CC) for that new treatment [23,24]. Figure 2 illustrates the difference between the CC, NCC, NCC concurrent observation time (NCC COT), and NCC non-concurrent observation time (NCC NCOT). The vertical dashed line in Figure 2 indicates the time that the new treatment B (the primary treatment of interest) is added. Patients randomized to the control group prior to the start of treatment B may have a period of observation time that overlaps with treatment B, which we call "NCC COT".

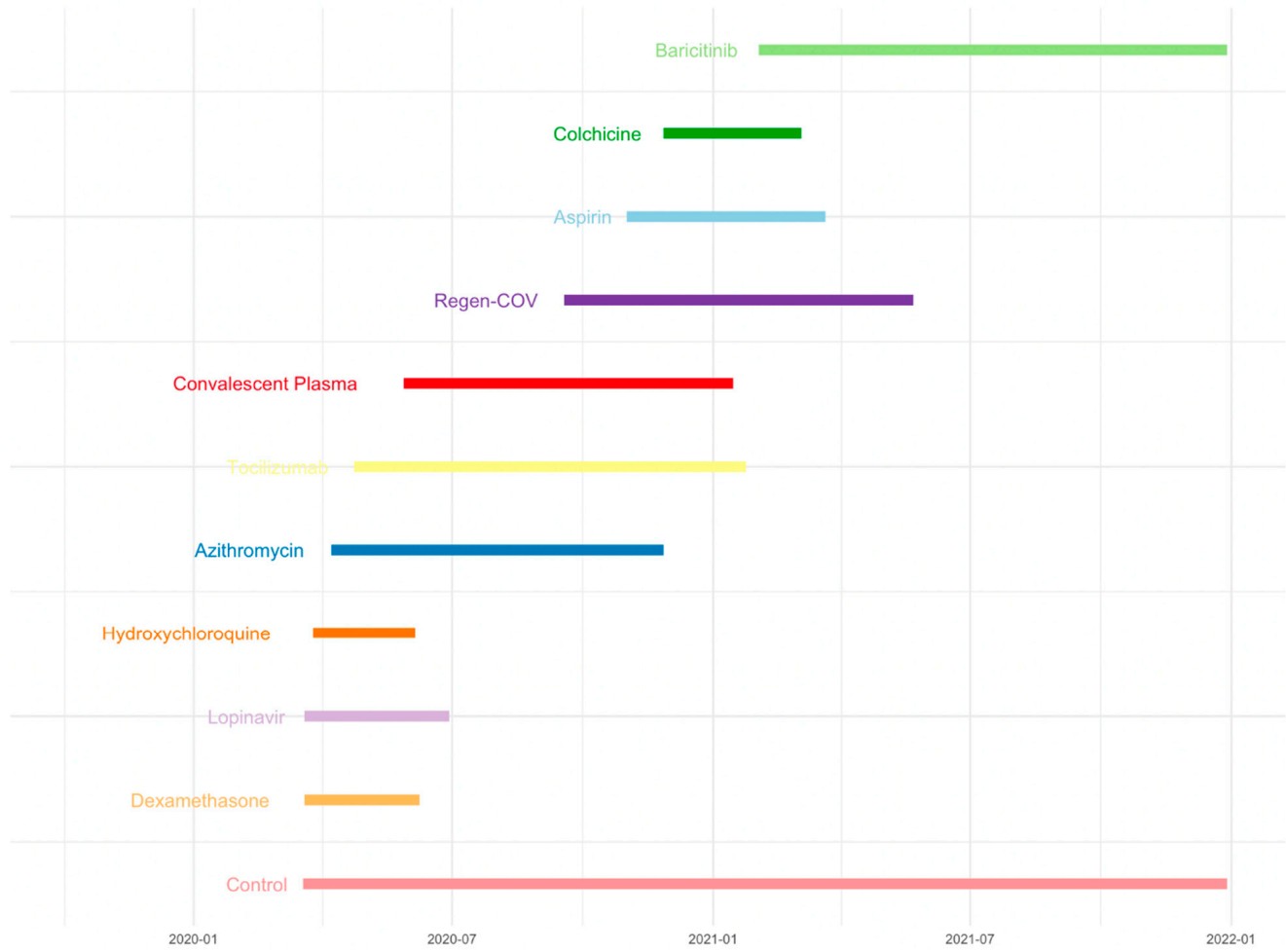

**Figure 1.** RECOVERY platform trial. RECOVERY is investigating whether treatment with Dexamethasone, Lopinavir-Ritonavir, Hydroxychloroquine, Azithromycin, Tocilizumab, Convalescent Plasma, Regen-COV, Aspirin, Colchicine, or Baricitinib prevents death in patients with COVID-19. In this trial, patients are randomly allocated between one or more treatment arms, each to be given in addition to the usual standard of care in the participating hospital. The study is dynamic, and treatments are added and removed as results and suitable treatments become available. Note that a common control group is shared for all treatments.

In this paper, considering a time-to-event endpoint under the proportional hazards model assumption, we propose an innovative approach using left truncation to borrow information from NCC COT, rather than considering only the CC group, or pooling CC with all NCC regardless of whether the observation time is concurrent or non-concurrent. Section 2 will introduce left truncation in detail, whereas Section 3 will present a simulated example to illustrate the performance of the approach. Section 4 will present a discussion on the assumptions and limitations of this approach.

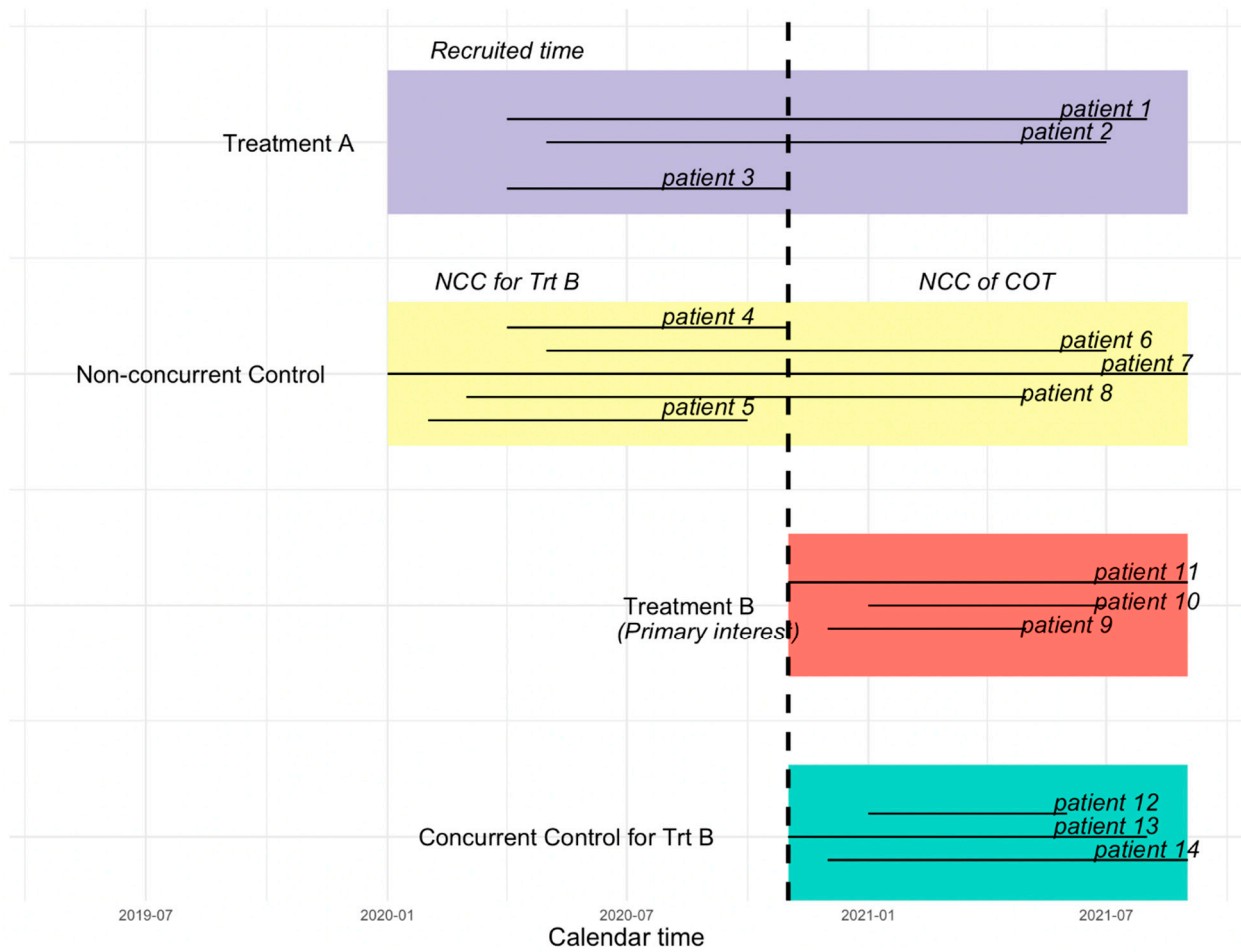

**Figure 2.** Non-concurrent control (NCC) and concurrent control (CC) in a platform trial. For simplicity, this platform trial only includes treatments A and B, with different entry times. Treatment A was investigated first, and then treatment B was added to the trial. The vertical dashed line represents the entry time of treatment B. Different color bars indicate different groups, and multiple lines within different color bars indicate different patients. For the evaluation of treatment B versus control, control patients recruited prior to the treatment B entry time are considered as non-concurrent control (yellow bar), whereas the control patients who are randomized concurrently with treatment B are considered as the concurrent control (CC) (green bar). Furthermore, within the non-concurrent control group, the observation time is divided into non-concurrent and concurrent observation time by the vertical dashed treatment B entry time.

## 2. Methodology

In this paper, we propose to borrow information from the concurrent observation time of NCC by left truncating the non-concurrent period, thus increasing the sample size of the control group and improving the statistical efficiency of active treatment–control comparisons. Here, left truncation refers to the approach that the survival time before a certain point is ignored, and only the survival time after a certain point is included in the analysis or the risk set [25]. For simplicity, we consider a platform trial that starts with treatment A and one control group, and a new treatment B is added during the conduct of the trial after the randomization of the initial treatment, which is illustrated in Figure 2. When treatment B joins and is deemed as the treatment of interest, the control group randomized after the new treatment joins is considered as the CC group, whereas the control group prior to the entry of treatment B is considered NCC for treatment B. In order to borrow information from NCC to increase statistical efficiency and reduce potential bias

introduced due to non-concurrent control, instead of borrowing the full information from NCC, we left truncate non-concurrent observation time.

By pooling the left truncated non-concurrent control concurrent observation time with the CC group, we consider fitting the classic Kaplan–Meier survival curve and Cox proportional hazards model. The Kaplan–Meier survival curve estimates the probability of survival over a certain period [26,27]. In a Cox proportional hazards model, we can specify a common baseline hazard function comparing the treated vs. CC, and the treated vs. NCC COT, and test if NCC COT and CC are different (either statistically or clinically). The Cox proportional hazards model is widely used in clinical studies to analyze both truncated and censored survival data [28,29]. The Cox proportional hazards model is defined as $\lambda(t|X) = \lambda_0(t)\exp(\beta X)$, where t represents the survival time, $\lambda_0(t)$ is the baseline hazard function, $\lambda(t|X)$ is a hazard function determined by a vector of covariates $X$, and $\beta = (\beta_1, \beta_2, \ldots, \beta_p)$ is a vector of regression coefficients, estimating the impact of covariates on hazard functions, and $\exp(\beta_i)$ represents the hazard ratio [30].

To incorporate left truncation in the Kaplan–Meier survival curves and the Cox proportional hazards model, we need to make some modifications to the risk sets for the left truncated data. Once we adjust the risk set incorporating left truncation, there is no requirement to include the left truncation time as a covariate for further adjustment [31]. Specifically, subjects who are left truncated at time L are not counted in the risk sets between disease onset time 0 and time L. They only appear in the risk sets after the left-truncation time L [32]. More specifically, suppose we have observed survival time $y_i$, with $j_i$ representing the index of the observation that has an event at time $t_i$, and $L_i$ representing the time for individual *i* enter the study. In the absence of left truncation, the risk set $R_i = \left\{ j : y_j \geq t_i \right\}$ assumes that every subject in the study is at risk of an event of interest at his or her time zero and continues to be at risk until the event occurs or when the subject is censored. When there is left truncation, individuals are not at risk before they enter the study. In the case of left truncation, the risk set becomes $R_i = \left\{ j : y_j \geq t_i > L_j \right\}$, where the time before the left truncation is not counted in the risk set, and only the time observed after the left truncation is considered in the risk set [32].

It is worth noting that to borrow NCC COT validly, we assume that the individuals in the CC group have the same hazard function as patients in the NCC COT after left truncation. In other words, we assume that the NCC subjects' prior experience in the non-current observation time does not affect their hazard functions in the current observation time. Though the assumption seems to be strong, it is the same assumption that is frequently used in any Cox proportional hazards model with a time-varying covariate. This approach also allows us to include any time-fixed covariates measured at baseline in the Cox model directly. However, if those baseline covariates were measured at time zero long before the truncation time for some NCC patients and can be extrapolated (such as age) or remeasured, one may use their covariate values corresponding to the truncation time as an alternative.

## 3. A Simulated Example

In this section, we present a simulated example to illustrate the approach. For simplicity, we assume the survival time follows an exponential distribution with scale parameter $\lambda$. In the non-concurrent control group, patients are recruited from $t_0 = 0$ to $t_1 = 3$ uniformly. The treatment and concurrent control group start at the time $t_1 = 3$ and patients are uniformly recruited until $t_2 = 6$. The maximum follow-up time for all three groups is $t_{max} = 15$. For the individual in the non-concurrent control group, we use the time-varying hazard function to generate survival time. We assume the baseline hazard $\lambda_0(t) = 0.2$ for $t\epsilon[t_0, t_1]$ and $\lambda(t) = 0.2exp(-0.4) = 0.134$ for $t > t_1$. Individuals in the CC group are assumed to have the same hazard function as NCC patients in the current observation time; thus, the scale parameter is $\lambda = 0.134$ in the CC group. For the treatment group, we assume $\lambda = 0.114$ to reflect the improvement in overall survival. Since we assume the same hazard function for the CC and the NCC COT groups, in Figure 3a, we pool the CC and NCC COT groups to obtain a pooled survival curve (blue curve). In Figure 3a, we can easily see

that the survival curves of the CC group (red curve) and the blue curve are similar. This implies that pooled survival curve is feasible and increases the sample size of the control group. Considering the 95% confidence interval for different groups, Figure 3a shows that there is a significant difference between the treatment and the pooled control groups, and Figure 3b shows that the 95% CI of the pooled control group is narrower than the CC group. Combining Figure 3a,b with Table 1, the standard error of the hazard ratio for the treatment group versus the pooled control group is 0.131, which is smaller than the hazard ratio for the treatment group versus the CC group (i.e., 0.147). Further, the *p*-value for the treatment group versus CC group comparison is 0.057, whereas that for the treatment group versus the pooled control group is 0.024, implying that significant differences can be found in the comparison of the treatment group versus the pooled control group when the hazard ratios are similar. Therefore, from the above-simulated example, we can conclude that extracting concurrent observation time from non-concurrent control can improve the efficiency of statistical inference in platform trials.

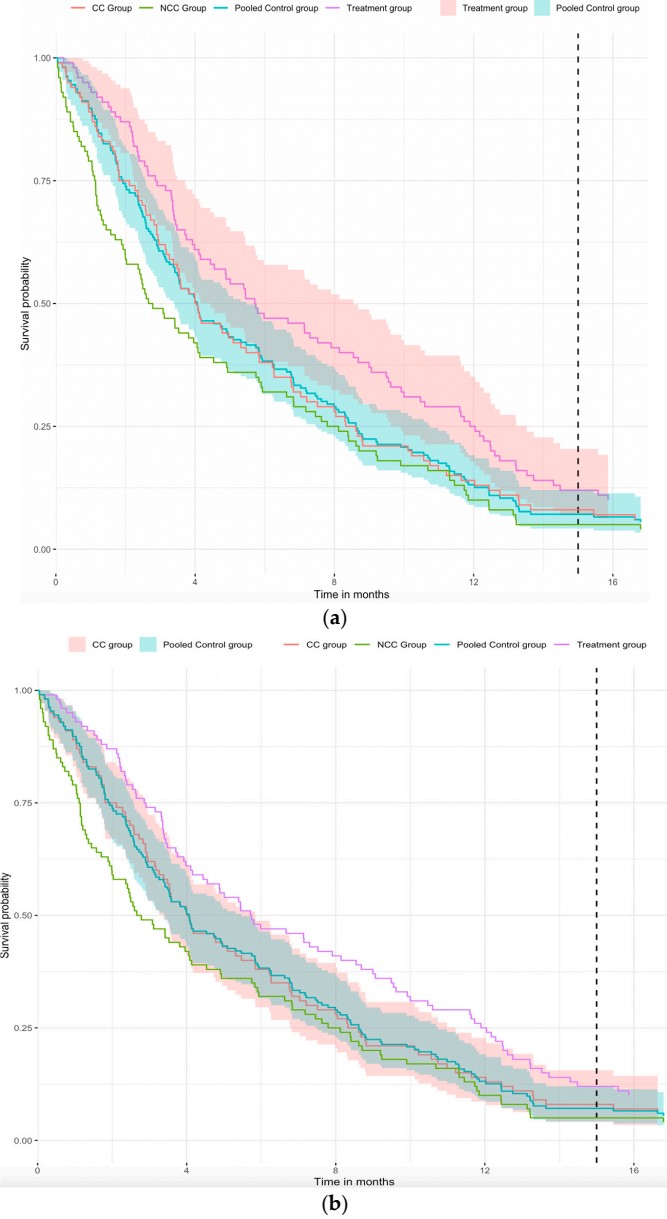

**Figure 3.** (**a**) Kaplan–Meier survival curves for different groups with 95% confidence intervals for the treatment and pooled control group. (**b**) Kaplan–Meier survival curves for different groups with 95% confidence intervals for the concurrent control and pooled control group.

**Table 1.** Estimated HR, SE, 95% CI, and *p*-values for Figure 3a,b.

| | True HR | Estimated HR | Estimated SE | 95% CI | *p*-Value |
|---|---|---|---|---|---|
| **Treatment vs. CC** | 0.850 | 0.755 | 0.147 | [0.566, 1.008] | 0.057 |
| **Treatment vs. Pooled Control** | 0.850 | 0.744 | 0.131 | [0.575, 0.962] | 0.024 |

In Figure 3a,b, the purple curve represents the treatment group, the green curve represents the non-concurrent control (NCC) group, the red curve represents the concurrent control (CC) group, and the blue curve represents the pooled control group of the CC group and NCC group with left truncation. The shaded portion of the figure represents the 95% confidence interval. In Figure 3a, the red-shaded part is the 95% CI of the treatment group, but in Figure 3b, the red-shaded part is the 95% CI of the CC group. The blue-shaded part is the 95% CI of the pooled control group in both figures. Note that we assumed the same hazard function of the CC group and the concurrent observation time in the NCC group.

## 4. Discussion

In this paper, we demonstrate that borrowing NCC COT through left truncating the NCC non-concurrent period in a platform trial can enhance statistical inference when the assumption of hazard functions holds in the Cox proportional hazards model and Kaplan–Meier curve estimation framework. This approach is closely related to the period analysis commonly used in cohort studies for estimating the survival functions within each calendar period. It involves dividing the study period into a set of non-overlapping calendar time intervals and analyzing the survival experience of individuals who are at risk during each calendar time interval separately. Subjects contribute their survival time to each period via left truncating the observation time before the beginning of the calendar period and right-censoring at the end of the calendar period [33]. For example, in the two large multi-center AIDS cohort studies, four sequential calendar periods corresponding to distinct therapeutic eras were used as instrumental variables to estimate the treatment effects of different HIV therapies [34,35]. In our approach, we borrow NCC COT through left truncating the non-concurrent period, which is the same idea as in the period analysis.

Generally, the concerns of directly borrowing all information from NCC come from two aspects, as summarized by Lu et al. [22]. First, the standard medical practice changes over time. For the same population, patients in the concurrent control group experiencing more advanced modern medical practice may perform better than patients experiencing less optimal medical practice. For example, for certain cancers, the community has seen patients on chemo treatment with improved responses over time, since the oncologists have accumulated extensive experience in selecting the right chemo for the patients and in managing the toxicity during the chemo treatment [36]. This concern is mostly relevant for some of the platform trials conducted over an extended period or those diseases in which the standard medical practices change rapidly. Our proposal of including concurrent observation time from NCC and truncating the non-concurrent observation time from NCC directly addresses this concern, assuming an NCC patient's prior experience during the non-concurrent observation period does not impact their future survival probability during the concurrent observation period. Second, the patient population drifts over time. Even though the NCC and CC are recruited under the same protocol with the same inclusion and exclusion criteria, the change in standard medical practice and other factors may impact the patient population. With this concern, one can include baseline covariates to potentially alleviate some population drift. Though it is always challenging to be certain about the inclusion of all confounding and prognostic factors to be adjusted, some statistical tools, such as inverse probability weighting, can be applied to handle the concern of population drifting [22]. Furthermore, one may consider Bayesian methods to dynamically down-weight the non-concurrent information from the non-concurrent control patients to further

improve statistical efficiency and potentially reduce sample size in the control group, as one can argue that additional useful information can be extracted from the non-current observation time. In addition, down-weighting non-concurrent information using power priors or commensurate priors may help to reduce the potential bias that may be introduced when combining non-concurrent and concurrent control patients in a single analysis.

When information from NCC is borrowed in a platform trial, it is generally more acceptable in exploratory studies rather than confirmatory studies, unless the recruitment of patients is extremely challenging, such as in the rare disease space (Project SignifiCanT: Statistics in Cancer Trials [37], the joint forum by FDA Oncology Center of Excellence, the American Statistical Association Biopharmaceutical Section Scientific Working Group, and LUNCGenvity Foundation [38]). As the proposed method has opened another possibility of borrowing NCC with potentially less bias when the assumptions hold, we advocate the approach to be considered in certain confirmatory settings with careful evaluations of the assumptions and early engagement of the regulatory agencies.

**Author Contributions:** Conceptualization, H.C. and C.L.; methodology, H.C.; software, J.L. and Z.J.; validation, J.L., H.C. and Z.J.; formal analysis, J.L.; investigation, C.L.; resources, C.L.; data curation, none; writing—original draft preparation, J.L.; writing—review and editing, J.L., C.L., Z.J., D.A., L.N. and H.C.; visualization, J.L.; supervision, H.C.; project administration, H.C.; funding acquisition, none. All authors have read and agreed to the published version of the manuscript.

**Funding:** This research received no external funding.

**Institutional Review Board Statement:** Not applicable.

**Informed Consent Statement:** Not applicable.

**Data Availability Statement:** The simulated data presented in this study are available on request from the corresponding author.

**Acknowledgments:** This article reflects the authors' views and should not be construed to represent the FDA's views or policies. We thank the anonymous reviewers for their thoughtful comments and constructive feedback to improve the quality of this work.

**Conflicts of Interest:** Lu is employed by AstraZeneca; Alemayehu and Chu are employed by Pfizer. They own stocks of their companies.

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
