# Peer review of "Borrowing Concurrent Information from Non-Concurrent Control to Enhance Statistical Efficiency in Platform Trials"

_curroncol, doi:10.3390/curroncol30040300_

Round 1

Reviewer 1 Report (Previous Reviewer 1)

The authors have adequately addressed comments raised in a previous round of review, and I feel that this manuscript is now acceptable for publication. 

This manuscript is a resubmission of an earlier submission. The following is a list of the peer review reports and author responses from that submission.

Round 1

Reviewer 1 Report

Overall, the article is well-written, the technique and process are solid, and the findings are accurate and intriguing. However, the only regret is that the authors gave only general applications rather than the applications of their main results.

Discuss in the context of previous work (with citations) how your results corroborate, contradict, or give new information. Concentrate the discussion on addressing the original aims and identifying the limits of the work and future study.

Please add a conclusion section in the manuscript that can address in at least three separate and brief paragraphs the following: i) main findings of the paper; ii) limitations of this work and future research; iii) broader impacts

Author Response

Please see the attachment. Below is a document containing all the revisions I have made for your reference. Thank you so much.

Reviewer 2 Report

This article presents an interesting area in the statistical method for that will increase the efficiency of platform trials. The method is motivated by the observation that the outcomes of the control arms in platform trials are often similar across different trials, and the authors propose using this information to inform the control arm for a new trial. The authors provide theoretical justification for their approach and demonstrate its performance through simulations and a case study using real data.

Here are some few comments:

1. The proposed method may be difficult to implement in practice, as it requires a large database of historical control arms to be available. 

2. The authors do not consider the potential biases that could arise from using information from non-concurrent control arms, such as differences in patient populations or treatments over time. The potential biases should be further addressed.

3. The simulation studies are limited in scope and do not consider all possible scenarios that could arise in practice. Therefore, the authors could discuss the practical implications of their method and provide guidance on how it could be implemented in practice, including recommendations for the size and scope of the historical control arm database that would be required.

Author Response

(The authors gave the same response as above.)
